# Blockchain Secured Dynamic Machine Learning Pipeline for Manufacturing

Fatemeh Stodt , Jan Stodt and Christoph Reich *

Institute for Data Science, Cloud Computing and IT Security, Furtwangen University of Applied Sciences, 78120 Furtwangen im Schwarzwald, Germany
* Correspondence: christoph.reich@hs-furtwangen.de

**Abstract:** ML-based applications already play an important role in factories in areas such as visual quality inspection, process optimization, and maintenance prediction and will become even more important in the future. For ML to be used in an industrial setting in a safe and effective way, the different steps needed to use ML must be put together in an ML pipeline. The development of ML pipelines is usually conducted by several and changing external stakeholders because they are very complex constructs, and confidence in their work is not always clear. Thus, end-to-end trust in the ML pipeline is not granted automatically. This is because the components and processes in ML pipelines are not transparent. This can also cause problems with certification in areas where safety is very important, such as the medical field, where procedures and their results must be recorded in detail. In addition, there are security challenges, such as attacks on the model and the ML pipeline, that are difficult to detect. This paper provides an overview of ML security challenges that can arise in production environments and presents a framework on how to address data security and transparency in ML pipelines. The framework is presented using visual quality inspection as an example. The presented framework provides: (a) a tamper-proof data history, which achieves accountability and supports quality audits; (b) an increase in trust by protocol for the used ML pipeline, by rating the experts and entities involved in the ML pipeline and certifying legitimacy for participation; and (c) certification of the pipeline infrastructure, the ML model, data collection, and labelling. After describing the details of the new approach, the mitigation of the previously described security attacks will be demonstrated, and a conclusion will be drawn.

**Keywords:** machine learning; verifiability; blockchain; cybersecurity





## 1. Introduction

Artificial intelligence (AI) and machine learning (ML) have emerged as key technologies in information security due to their ability to rapidly analyse millions of events and identify a wide range of threats. Blockchain is distinguished by its advantages of decentralised data storage and program execution, as well as the immutability of the data. Blockchain and AI can work together to provide secure data storage and data sharing, and to analyse the blockchain audit trail to more accurately understand relationships of data changes. Combining the two technologies will give blockchain-based business networks a new level of intelligence by allowing them to read, understand, and connect data at lightning speed [1].

ML has been established as the most promising way to learn patterns from data. Its applications can be found in the data processing of web browsing, financial data, health care, autonomous automobiles, and almost every other data-driven industry around us [2]. Because of the vast range of applications, ML models are now being run on a wide range of devices, from low-end IoT and mobile devices to high-performance clouds and data centres, to offer both training and inference services [3].

Especially in industrial production, machine downtimes are costly for companies and can have a significant impact on the entire company [4]. Therefore, AI applications used in the business world must be fail-safe, secure, and reliable. Trust in AI applications that are used in industrial production can only be built if they meet the requirements for information security [5]. Next to the typical cybersecurity threats, the ML-based threats must be added to these requirements, which are threats against ML data, ML models, the entire ML pipelines, etc. [6]. To build trust, all actors must make sure they are following the rules, and their actions must be verifiable and believable to their partners. For accountability, the goals of security protection are for the assets to be available, safe, private, and handled in a way that obeys the law (e.g., privacy).

Machine learning applications are, in fact, pipelines that link several parts and rely on large amounts of data for training and testing [7]. Data are also required for maintaining and upgrading machine learning models, since they take user data as input and use it to come up with insights. The importance of data in machine learning cannot be overstated, as data flows across the whole machine learning process.

Widespread attacks threaten ML security and privacy, these range from ML model stealing [8], "model inversion" [9], "model poisoning" [10], "data poisoning" [11], "data inference" [12] to "membership/attribute inference" [13] as well as other attacks. In ML, security and privacy problems are caused by complex pipelines that use multiple system and software stacks to offer current features such as acceleration. A full ML pipeline includes collecting raw data, training, inference, prediction, and possibly retraining and reusing the ML model. The pipeline may be segmented since data owners, ML computation hosts, model owners, and output recipients are likely separate businesses [14]. As a result, ML models frequently have weak resilience, as shown in adversarial cases or poisoning assaults. A small change in the way training is conducted could have huge negative effects that are hard to spot.

In comparison, blockchain has an entirely different purpose and characteristics. Blockchain is a technology to store data, immutable and decentralised. The data are distributed across a large network of nodes so that it is available even if some nodes fail. Once a block has been added by agreement among participants, it cannot be deleted or changed, even by the original authors. The data are publicly available but not publicly readable without a digital key [15]. One obvious use is to save records of success and credit, such as any entity credentials and trust amounts related to it [16]. The granting institution would upload the certificate data to the blockchain, which the member may view or connect from web pages [17].

Some research has looked at solutions for attacks on ML models or pipelines [14], but these solutions are applied to the central system. But as communication and distributed systems improve, different companies or developers can now work on platforms that use distributed systems. However, the question is how they can trust each other's output when there is no central authority.

In this study, we present a blockchain-based framework to help find answers to ML's security, traceability, and privacy problems. In manufacturing, this framework is used to keep data private and accessible for quality control and to explain those ideas. The proposed solution has several key benefits, such as a data history that can't be changed, which provides accountability and makes quality checks possible. By making a protocol for the ML pipeline that is used, rating the experts who are part of the ML pipeline, and making sure that they are legitimate, trust is increased. In addition, pipeline architecture and the machine learning with its tasks of model data gathering, and labelling are certified. After the details of the framework are explained, it will be shown how to stop the security attacks that have already been made public.

The article is structured as follows: Section 2 describes the utilised research methodology. The method follows the "Systematic Literature Review" to detect research gaps and research objectives. Section 3 describes the current state of the art in focusing on ML pipelines and how blockchain could help secure ML use cases. Section 4 discusses the security challenges of machine learning models and pipelines, which frequently have the potential to destroy the output result of the model. In Section 5, we describe the new

approach to maintain the security of the model. For this purpose, certificates are introduced which can be evaluated by a trust mechanism within the blockchain. Section 6 explains a typical use case of visual quality inspection. We provide security analyses and evaluation in Section 7 and the last section, Section 8, concludes the paper.

## 2. Research Methodology

This section introduces the research methodology used. The method follows the "Systematic Literature Review" method developed by Okoli and Schabram [18], and adapted by Dr. Heil of the Justus Liebig University of Giessen [19]. The steps are as follows:

1.  Definition of literature research question and research objective: Definition of research questions, research principle and naming the target of the research
2.  Inclusion and exclusion criteria: Define inclusion and exclusion criteria, documentation of refinements and changes
3.  Databases: Determination of databases/search engines
4.  Define search components: Definition of search terms, scheme for search term entry, search for synonyms of search terms
5.  Define search strings: Developing search strings with search components, search terms, synonyms and operations, describing the search procedure, checking the search strings using the PRESS checklist
6.  Conducting the research: Input of search strings, documentation of changes, documentation of results, application of solution suggestions in case of too few or too many hits

In the following, these steps are worked through.

### 2.1. Step 1. Define Literature Research Question and Research Objective

Four literature research questions are to be answered; these are:

*   RQ1: What is the definition of an ML pipeline?
*   RQ2: What are the benefits of an ML pipeline?
*   RQ3: What are security risks of an ML pipeline?
*   RQ4: Which of the identified risks can be addressed by blockchain?
*   RQ5: How can blockchain and ML pipelines be linked together?

The literature review must cover the domains of ML pipeline, its risks and benefits, how these risks can be addressed by blockchain and the combination of blockchain and ML pipelines. The literature review must be a complete (sensitive) analysis in order to answer the research questions posed. It is expected that there will be surveys covering this. The methods mentioned in these surveys should also be considered.

### 2.2. Step 2. Inclusion and Exclusion Criteria

Table 1 shows the inclusion/exclusion criteria for each of the four examined domains.

The literature found should fall within the scope of production. In general, literature that is not available as full text (via library access Furtwangen University) is excluded.

**Table 1.** Inclusion/Exclusion Criteria.

| Inclusion/Exclusion | Domain | Criteria |
| --- | --- | --- |
| Inclusion | ML Pipeline | Definition<br>Benefit<br>Risk |
| Exclusion | | Case Study<br>Use Case |
| Inclusion | Blockchain | Addressing security risks |
| Exclusion | | Security risks of blockchain itself |
| Inclusion | Combination of ML Pipeline and Blockchain | Approaches of combining the benefits of ML Pipeline and Blockchain |
| Exclusion | | Any form of performance improvement |

### 2.3. Step 3. Determination of Databases/Search Engines

Google Scholar is selected as the search engine of choice. This is because Google Scholar indexes a wide range of high quality journals (IEEE, ACM, Elsevier, Springer, etc.) as well as the preprint service arXiv, where numerous papers have been pre printed over the years that were later pre-reviewed in one of the before mentioned high quality journals. In addition, a multidisciplinary comparison of coverage by citations by Martín-Martín et al. [20] has shown that Google Scholar is the most comprehensive data source (Studied data sources: Google Scholar, Microsoft Academic, Scopus, Dimensions, Web of Science, and OpenCitations' COCI).

### 2.4. Step 4. Define Search Components

The keywords are defined for the three domains: "ML Pipeline" (Table 2), "Blockchain" (Table 3) and "Combination of ML Pipeline and Blockchain" (Table 4), as well as the combination of keywords per domain. The keyword "survey" is used in each of the 3 domains to find survey papers to uncover literature that may have been excluded by the chosen keywords. The aim of this is to create a complete literature review.

**Table 2.** Literature Review Keywords—ML Pipeline.

| Keyword Component 1 | Keyword Component 2 |
| --- | --- |
| ML Pipeline | Definition<br>Benefit<br>Risk |

**Table 3.** Literature Review Keywords—Blockchain.

| Keyword Component 1 | Keyword Component 2 |
| --- | --- |
| Blockchain | Addressing security risks<br>Addressing risks |

**Table 4.** Literature Review Keywords—Combination of ML Pipeline and Blockchain.

| Keyword Component 1 | Keyword Component 2 |
| --- | --- |
| Combination of ML Pipeline and Blockchain | Combining the benefits of ML Pipeline and Blockchain |

### 2.5. Step 5. Define Search Strings

The search terms listed below in Tables 5–7 are used to search for literature in Google Scholar for the three categories mentioned and are represented using logic symbols (OR: $\vee$, AND: $\wedge$). It should be noted that search string expressions could have been further simplified, but this was omitted for the sake of clarity. The Peer Review of Electronic Search Strategies (PRESS) [21] is used to detect errors in the search strings that could complicate or compromise the search.

**Table 5.** Literature Review Search Strings—ML Pipeline.

| Search Strings |
| --- |
| Machine Learning Pipeline ∨ ML Pipeline ∧ (Definition ∨ Benefit ∨ Risk) ∨ Survey |

**Table 6.** Literature Review Search Strings—Blockchain.

| Search Strings |
| --- |
| Blockchain ∧ (Addressing security risk ∨ Addressing risks) ∨ Survey |

**Table 7.** Literature Review Search Strings—Combination of ML Pipeline and Blockchain.

| Search Strings |
| --- |
| Combination of ML Pipeline and Blockchain ∨ Combining the benefits of ML Pipeline and Blockchain |

*2.6. Step 6. Conduct the Research*

The following documents the conducted literature review using the "Preferred reporting items for systematic reviews and meta-analyses" (PRISMA) [22]. For the domain "ML Pipeline" five papers were found, "Blockchain" three papers were found, and for "Combination of ML Pipeline and Blockchain" six papers were found.

## 3. State of the Art

A machine learning pipeline is an approach to making machine learning models and all the processes behind them more productive. The pipeline is for large-scale learning environments that can store and work on data or models better with data parallelism or model parallelism [23]. As a backbone for distributed processing, ML Pipeline has a lot of benefits, such as being easy to scale and letting to debug data distribution in ways that local computers cannot [24,25].

A significant point in the design and implementation of the ML pipeline is the ability to use the ML models in manufacturing [26]. It is important to pay attention to the sensitivity and security of the ML pipeline in industry. ML-based communications and networking systems demand security and privacy. Most ML systems have a centralised architecture that is prone to hacking since a malicious node only has to access one system to modify instructions. Training data typically incorporates personal information, and therefore data breaches may affect privacy. Hackers must be kept away from ML training data [14].

When training an ML model, a lot of data from many different places is often needed, which raises privacy concerns. To prevent identity exposure, each node in a blockchain system communicates using a created pseudonymous address. By using pseudonyms, blockchain may offer pseudonymity and be acceptable for specific use cases that demand strong privacy [27]. Furthermore, the privacy of data/model owners is protected by cryptographic techniques, and the confidentiality of data/model sharing across numerous service providers is assured [28].

Blockchain qualities such as decentralisation, immutability, and transparency open up new opportunities for ML algorithms employed in communications and networking systems [29–31]. Discuss blockchain for ML in this area, including data and model sharing, security and privacy, decentralised intelligence, and trustful decision-making.

Ref. [32] envisions a permissionless blockchain-based marketplace for ML professionals to acquire or rent high-quality data. As part of sharding, the network is split into several Interest Groups (IG) to make data exchange more scalable. Users are encouraged to gather helpful knowledge regarding a subject of interest to them. Each IG has its own dataset that incorporates data from all of its nodes. IG members might be recognised for the quantity and quality of their data. Ref. [33] presents ADVOCATE to manage personal data in IoT situations. The proposed framework collects and analyses policy data to make decisions and produce user-centric ML solutions. Blockchain technology is used in the suggested architecture so that data controllers and processors can handle data in a way that is clear

and can be checked. All consents would be digitally signed by the parties to the contract to make sure they couldn't be revoked, and the hashed version would be sent to a blockchain infrastructure to protect the data's integrity and keep users' identities secret.

On the other hand, data dependability is crucial to ML algorithms. For ML approaches to solve problems more effectively, they need more data sources to train their models on during the analysis of data resources. However, in today's sophisticated and trustless networks, the goal of high accuracy and privacy-aware data sharing for ML algorithms remains problematic. Because of privacy and reputation concerns, most users are hesitant to share their data with the public.

The authors of [34] describe a crowdsourced blockchain-based solution to enable decentralised ML without a trusted third party. A non-cooperative game theoretic strategy with two workers auditing each other and a cryptographic commitment instrument are presented to tackle employee interaction (blockchain nodes) and free-riding difficulties in crowdsourcing systems. The expensive and randomised computation is crowdsourced via the application layer and "asynchronously" executed. Full nodes/miners may insert the output into the next block as soon as the result is submitted, rather than waiting for mining to finish.

In [35], the authors suggest a reputation-based worker selection strategy for assessing the dependability and trustworthiness of mobile devices in mobile networks. They employ a multi-weight subjective logic model and consortium blockchain to store and maintain worker reputation in a decentralised way in order to deliver trustworthy federated learning. For collaborative federated learning, to enable mobile devices to share high-quality data, we present an effective incentive system that combines reputation with contract theory.

The study discussed above has the potential to serve as the foundation for developing decentralised, transparent, secure, and trustworthy ML-based communications and networking systems. They are still being discussed based on plausible ideas, but they are still in their early stages. Some technical concerns, such as scalability and incentive issues, need more research.

## 4. Security Challenge of ML Models and ML Pipelines

To achieve trust, security issues have to be considered across the whole pipeline. We divide current attack vectors into availability, confidentiality, integrity, and accountability, which are the most important parts of information security. Every vulnerability in the ML pipeline may be attacked, and this section can only give an overview of the countless attacks.

### 4.1. Attacks against Availability

Attacks against the availability should not be underestimated since the pipeline is complex and spread over several providers. The attack surface is therefore huge, and attackers steal hardware, crash software, and use denial of service to reduce network bandwidth or utilize services [36].

### 4.2. Attacks against Confidentiality

Attacks against confidentiality may happen on sensitive data and model information while honestly executing ML training and inference without altering the computation outcomes. Because the data and models are accessible by several stakeholders, these attack vectors often occur in the pipeline phases of model training, model deployment or inference, or model upgrades [37].

In federated ML, the host that orchestrates all clients' local training may access their updated models and use these models to infer private information about their local data [38]. Among the most common attacks are: data reconstruction attacks [39], which aim to reconstruct original input data based on the observed model or its gradients; attribute inference attacks [40], which aim to infer the value of users' private properties in the training data, and membership inference attacks [41], which aim to learn whether specific data instances are present in the training dataset.

### 4.3. Attacks against Integrity

An attacker might intentionally undermine the ML's integrity by exploiting training or inference results. An integrity attack is a data corruption effort. It's usually a planned attack by malware that deletes or changes the information in a dataset. It also involves attackers encrypting sensitive or critical data. Previous research has demonstrated that model accuracy may be degraded by just compromising thread scheduling in a multi-threaded ML pipeline [42].

Another example is a model poisoning attack. In order to obtain an alternative model decision, an attacker lowers the model's performance [43]. Poisoning a model involves replacing a good one with a bad one. In a traditional cyber attack, this is relatively simple. A model, once trained, is nothing more than a file on a computer, similar to a picture or a PDF document. Attackers may breach the systems that hold these models, then edit or replace the model file with a corrupted one. Even if a model has been trained on an uncorrupted dataset, it may be substituted by a corrupted model at different points of the distribution pipeline. Furthermore, the training set may be manipulated, for example, by including data with calibrated noisy labels [44]. This causes the classifier to have incorrect bounds for certain data points.

### 4.4. Attacks against Accountability

Another common method for categorising ML attack surfaces is whether an attack needs access to the internal architecture of an ML model [45]. In black-box attacks, model theft usually starts from the outside, with no prior knowledge, and the goal is to learn the model itself. Membership inference attacks on data privacy are often black-box attacks because they are much more effective than white-box attacks. White-box access has no discernible effect on the attack's "advantage" in revealing membership privacy. White-box attacks include almost all data reconstruction attacks, some adversarial example attacks, attribute inference attacks, and so on. An overview of security challenges in industrial machine learning pipelines is shown in Table 8.

**Table 8.** Overview of security challenges in industrial machine learning pipelines.

| Attack Against | Attack Type | Caused Phase of ML | Impact | Reference |
|---|---|---|---|---|
| Availability | Dos Attacks | ML model | Triggers a buffer overflow in image processing | [36] |
| Confidentiality | Data Reconstruction Attacks | ML training | Modify data | [11,37] |
| Confidentiality | Attribute Inference Attacks | ML training | Inferring the value of users' private properties | [46] |
| Confidentiality | Membership Inference Attacks | ML training | Learning specific data | [13] |
| Integrity | Adds calibrated noises | Data Collection, ML training | Exploit training/inference result | [42–44,47] |
| Accountability | Model stealing | ML model | Learn the model | [45,48] |

## 5. Proposed Method for Mitigation

### 5.1. System Overview

The proposed system is a machine learning pipeline built on a private blockchain using InterPlanetary File System (IPFS) [49] for a distributed storage. As indicated in Section 4, there are several security and trust issues in the machine learning pipeline that can cause unpredictable progress. The collaboration and tracking of the action of each participant in the pipeline, the stages of transparent ML progress given by the blockchain, enable the establishment of trust and assurance of the quality of the pipeline's output. To ensure data protection, the entities in this system are located in a private blockchain, where the participants are registered and certified members. The trust manager nodes, which review certificates and adjust stakeholders' trust levels based on their actions. Because the ML pipeline is safe, stakeholders can request its development so that pipeline stakeholders can monitor tracing and exchanging information.

The proposed system is shown in Figure 1 and will be discussed in detail in the next subsections.

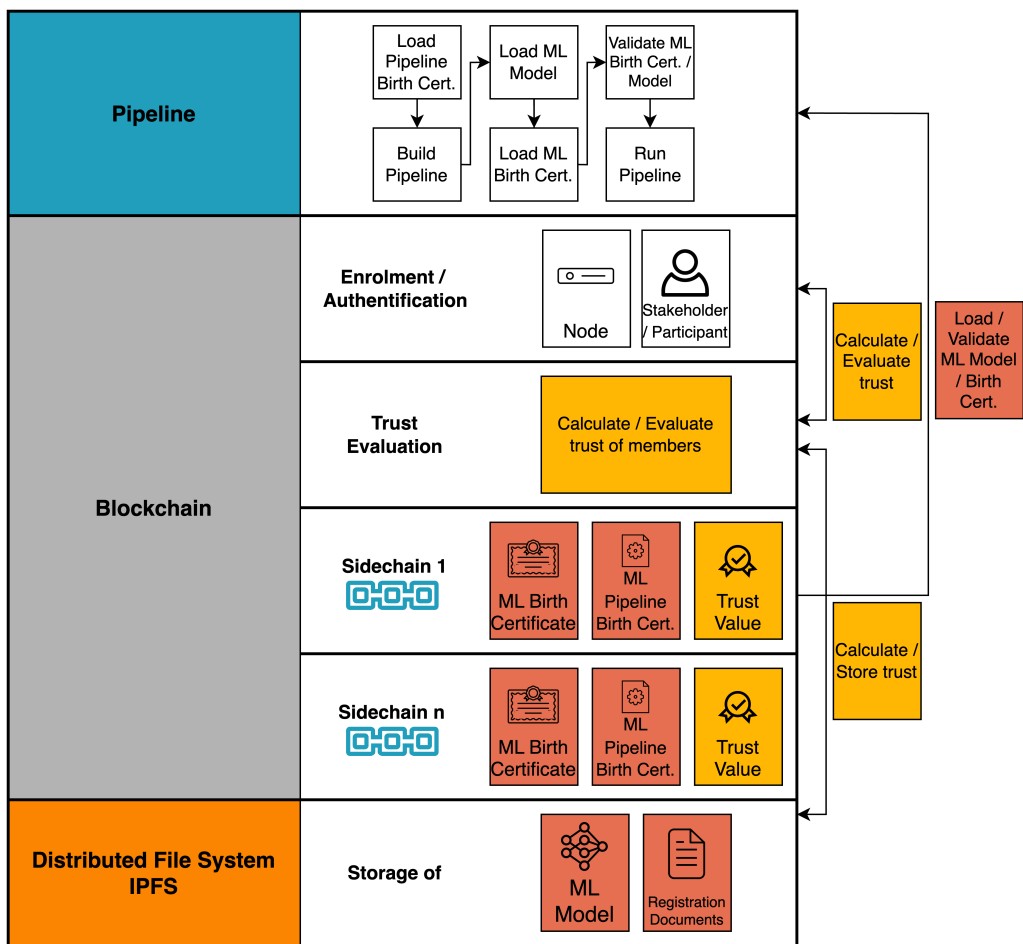

**Figure 1.** System overview.

## 5.2. Blockchain Infrastructure

In this proposed architecture, a private blockchain with participants of the ML pipeline are offered to host their own blockchain node. It is shown in Figure 2. Blockchain members can include the owner of a factory or company, the IT department, machine learning specialists, the consultant share for each project, employees from each area who need to prepare data, and data analysts. Authorisers from each area can also be blockchain members. Each participant in this scenario must be a member of the Blockchain, as in any other real-world initiative. Some authoritative organisations may have permanent members who specialise in blockchain or machine learning. Each member must be certified by the blockchain authority and have a trust value as feedback from the blockchain about its activities.

Clients, IT consultancies, data providers, data analysts and government agencies are all examples of nodes in this blockchain (see Figure 2). Each node may have a different level of trust depending on its previous activity, but all must be accredited. The most trusted nodes can be part of the blockchain committee that approves new nodes and validates and creates blocks in the ledger. Smart contracts can also be used by nodes to set up sidechains and send certain information to the main chain. The blockchain infrastructure is the main trust anchor of the total system.

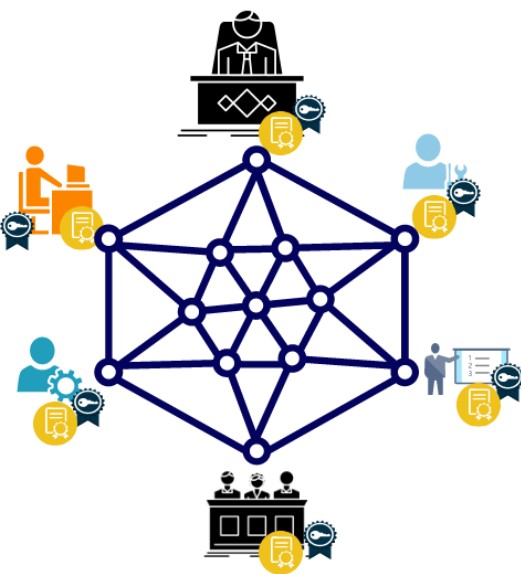

**Figure 2.** Private Blockchain.

*5.3. Authenticate and Enrolment*

Due to the importance of model security and privacy when accessing and analysing datasets, only certified and authorised individuals should have access to the data and participate in the process of building machine learning models. It is therefore recommended to use a private blockchain to regulate the registration of new nodes and to monitor existing blockchain nodes. Current nodes cannot provide blockchain services in this case, if they do not have a valid certificate and a sufficiently high trust level in their profiles. Depending on the application driven security level, the attributes, that are part of the certification, varied. Here are some examples of possible certificates and their attributes, (see Table 9).

**Table 9.** Certification entities and their possible attributes.

| Certification Entity | Possible Attributes |
|---|---|
| Companies | Legal entity of the participating company |
| Customer | Identity of the person, Affiliation, Company certificate |
| Domain expert, Data scientist | Identity of the person, Affiliation, Company certificate Expertise degree |
| ML model | A detailed description can be found in Section 5.4.1 |
| Data sources Data | Identity of the source Data sources, Type of data, Data boundaries (e.g., min, max) |
| Public data set | Source, Signature (Hash) |

Enrolment of New Nodes

If a new stakeholder/participant wants to join the ML pipeline and additional nodes are required to join the blockchain, the new node can request registration by executing a smart contract that contains the required information (e.g., certificate) and links to the documents uploaded by the IPFS. If the information in the received enrol transaction is correct, the committer issues a certificate for the node and adds the node ID to the associated group of the node role. This can be automatically added in case of non-critical participations (e.g., viewing ML results) or can be with further interactions (data owner has to give access permission, . . . ).

### 5.4. Sidechain

For each new pipeline project, the primary stakeholder should execute a smart contract to create a sidechain and add the remaining stakeholders if they are blockchain members, or enrol them if they are not. This ensures isolation in a multi tenant environment.

In the next step, they allocate an IPFS environment to share the data between sidechain members. Each member can store data in a specific IPFS and sign data with his blockchain signature. Each transaction submitted by a member involves an IPFS link that contains data files. After review of the transactions by members, a block is created inside the sidechain that contains the transactions, feedback, and trust score for that transaction.

All generated information that needs to be certified, such as the model's birth certificate (see Section 5.4.1), is saved in the sidechain as a local certificate. Figure 3 shows one sidechain instantiated by the main chain and depicted a second sidechain to show the multi tenant feature of the main chain. After completing the project, depending on the level of privacy, it can be sent to the main blockchain as shows in Figure 3 and after being reviewed by experts and receiving their trust, the local certificate can be upgraded to blockchain certificate level.

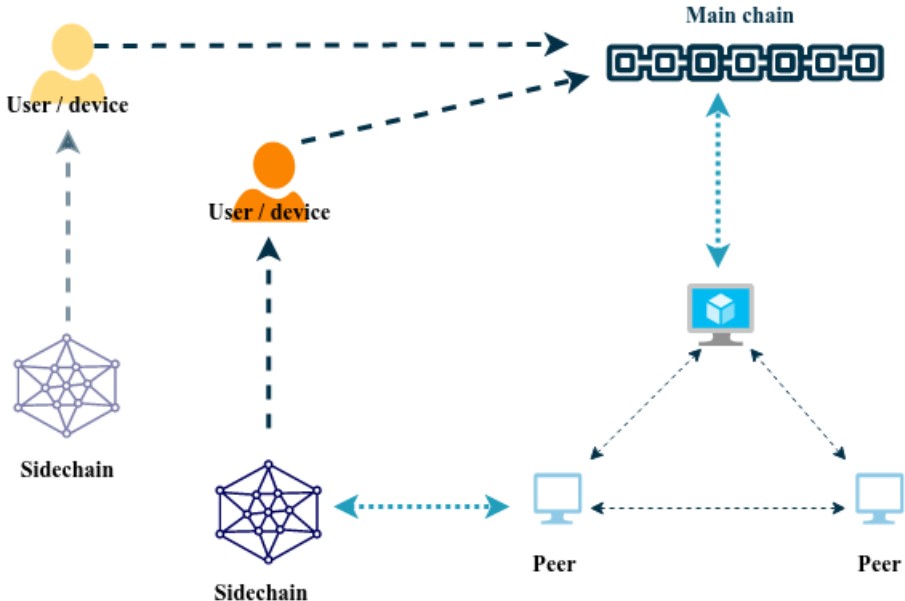

**Figure 3.** Sidechain Process.

### 5.4.1. Create Birth Certificates

After each stakeholder within the blockchain has been made traceable through certificates and trust management, two further important components of this machine learning pipeline need to be secured. These two components are the ML models and their creation, and the machine learning pipeline (components, software, configurations) itself. The aim is to make all software and its configuration involved in the ML pipeline traceable.

### Machine Learning Model Birth Certificates

The Machine Learning Model Birth Certificate stores all relevant information that allows conclusions to be drawn about its creation. These are, as described in more detail in a previous paper [16], information about the model life cycle phases (1. Model Requirements, 2. Data Collection, 3. Data Cleaning, 4. Data Labeling, 5. Feature Engineering, 6. Model Training, 7. Model Evaluation, 8. Model Deployment, 9. Model Monitoring) [50].

The most important information for each phase are presented below:

1.  Model Requirements: Goal of the model, decision made to achieve the model goal.

2. Data Collection: Information about data acquisition, sensor model, environment parameter.
3. Data Cleaning: Description of the cleaning method, parameters and software.
4. Data Labelling: Description of the labelling method, instructions, software and person who carried out the labelling.
5. Feature Engineering: Method and parameters of data augmentation.
6. Model Training: Hyperparameters, etc.
7. Model Evaluation: Accuracy, precision, etc.
8. Model Deployment: Information about machine learning pipeline (described in more detail later).
9. Model Monitoring: Alert threshold.

These machine learning model birth certificates are stored in the blockchain and contain a reference to the certificate of the creator of each phase.

Machine Learning Pipeline Birth Certificates

As with machine learning model birth certificates, a birth certificate is created for the machine learning pipeline that includes the architecture, the structure and processes within the individual components, software versions and configurations to make it traceable; the selection of this information is justified below.

- Architecture:
    - The architecture of the ML pipeline determines the order of execution of the individual components. Since the components change the state of the data, a change in the order results in a different outcome.
    - Certain pipeline architectures are to be considered invalid. Determined results are therefore also to be considered invalid.
- Components:
    - As with the entire ML pipeline, the order in which the subcomponents are executed must also be known in the components, since a change in order changes the state of the data.
    - Since the components change the state of the data, it must be known which processes take place in these components.
- Software versions:
    - It is important for the traceability in case of errors in the ML pipeline to be able to determine afterwards which software versions were used.
    - If it is known which software versions are used in the ML pipeline, it is possible to react more quickly to known errors in the software by updating.
- Configurations:
    - Software usually has configuration options. If this configuration is changed from the standard configuration, unintentional errors may occur in certain cases.
    - Saving the software versions can be used not only to find errors, but also to quickly update the configuration in case of known errors.

*5.5. Pipeline*

After the architecture of the ML Pipeline is described by the combination of the individual components by the respective stakeholders in the ML Pipeline Birth Certificate, which is stored in the project specific sidechain, the architecture is built by the stakeholders. For this purpose, the configuration is used for each of the components and the necessary software is installed with the corresponding version and configuration. For the components of the ML usage, the corresponding ML birth certificate is loaded from the project specific sidechain and the ML model described there is retrieved and loaded into the inference engine. Once the ML pipeline is built, it can be used for the intended use case.

*5.6. Trust Evaluation*

The trust management system is embedded in the blockchain and defined by a series of smart contracts to obtain the trust score of each member as a local reputation system, calculate trust value, and update the trust table of members inside the main chain.

5.6.1. Trust Value

Trust evaluation is the process of quantifying trust with attributes that influence trust. It is a challenging process, because of lack of essential evaluation data, demand of big data process, request of simple trust relationship expression, and expectation of automation. Mostly, the essential trust attributes are use case dependent. Assuming the use case (see Section 6), these attributes could be:

- Human ML pipeline participants, e.g., work experience: the more the more trust
- ML pipeline infrastructure, e.g., software versions: the higher the more trust
- Data source dependent, e.g., age of the machine: the older the less trust
- etc.

Above show some examples a trust value can be calculated. Our approach simplifies it by categorizing the trust value into 7 level, as see in Table 10.

**Table 10.** Possible Trust Values for Entities.

| Trust Values | Label |
|---|---|
| >0.75 to 1.0 | Very High Trust |
| >0.5 to 0.75 | High Trust |
| >0.25 to 0.5 | Medium Trust |
| >−0.25 to 0.25 | Low Trust/Distrust |
| >−0.5 to −0.25 | Medium Distrust |
| >−0.75 to −0.5 | High Distrust |
| −1.0 to −0.75 | Very Distrust |

The determination of trust can be modelled more precisely, if you use more historic data of the behavior of the trust entity. The paper [51] gives an overview of trust evaluation using machine learning for predicting the trust.

5.6.2. Four Steps of Trust Management

The procedure consists of four steps:

1. Each member inside the sidechain should, by submitting the transaction, send a request for feedback on the transaction. We name it an "attached transaction" because the feedback to it will be sent by the main transaction number. Other stakeholders can provide feedback and a trust score for the transaction after it has been reviewed and investigated. After mining, the block transactions and attached transactions are immutably stored in the sidechain.
2. The blockchain executes a smart contract after a specific time per day to collect local trust values from sidechains.
3. After getting the request for trust value, the sidechain executes a smart contract to gather and calculate trust scores saved in blocks and submit the trust table to the main chain as a transaction.
4. After receiving transactions from all sidechains, the results calculate and update the trust value table inside the main chain.

*5.7. Storage Certificates and Trust Level*

During the pipeline project, each certificate related to the pipeline, the ML model's birth certificate, the raw data certificate, and the dataset certificate are stored in a sidechain. In parallel, certificates for new members, such as experts or businesses, are being stored in

the main chain. In addition, the trust value, which is calculated in Section 5.6 is updated and stored in the main blockchain. If approved by the rest of the blockchain experts, each of the local certificates can be stored in the main blockchain and upgraded to a global certificate.

## 6. Case Study: Machine Learning Application in Manufacturing

This section talks about a typical application of machine learning in a manufacturing SME, as well as the ML pipeline and the people who are involved.

### 6.1. Use Case: Visual Metal Surface Quality Inspection

As an example use case for the presented concept of the "Blockchain Secured Dynamic Machine Learning Pipeline" (Blockchain Secured Dynamic Machine Learning Pipeline (bcmlp)), the application of machine learning for visual metal surface quality inspection is presented, as seen in Figure 4.

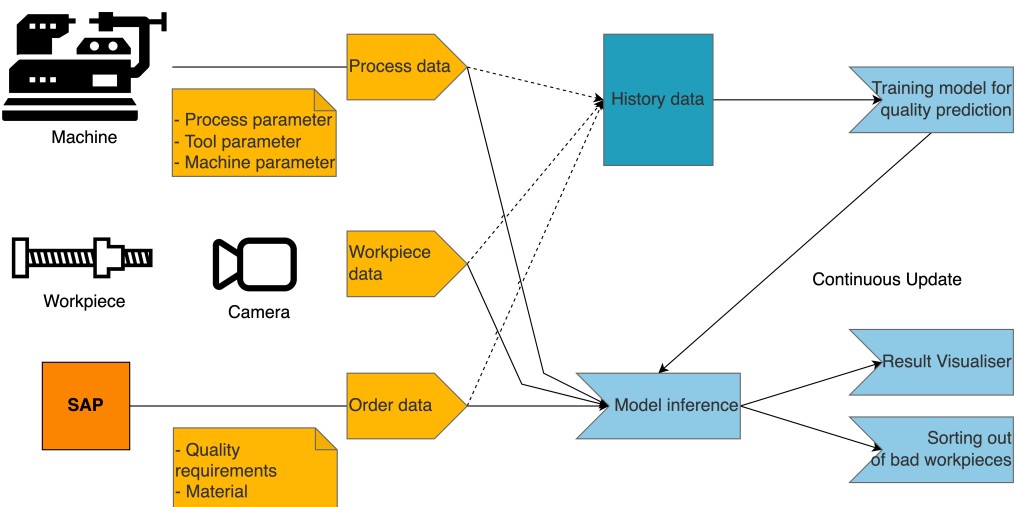

**Figure 4.** Visual Metal Surface Quality Inspection.

Figure 4 shows the input parameters that influence the quality inspection process for the machine learning model. They are: process data (process parameter, tool parameter) of the *machine*, *workpiece* data (surface image), *order* data (quality requirements, material, etc.). From the collected data in the *history data base*, a data scientist trains a machine learning model, which afterwards is used for *model inference*. The model is always being trained and updated to take into account changes in parameters and the quality changes that come from them. The results of the *model inference* are visualised for the machine operator, and workpieces that do not meet the required quality requirements are sorted out.

### 6.2. ML Pipeline and Stakeholders

Typically, a machine learning pipeline consists of five components that build on each other, are continuously monitored by a monitoring system, and are based on CRISP-DM (CRoss-Industry Standard Process for Data Mining) [52]. Through a series of intermediate steps and components, the raw data that comes in (process parameters, workpiece data, and order data) is turned into a quality index.

Step 1 "Data collection" (see Figure 5), where the data to be processed is collected, is the entry point into the ML pipeline. Step 2 "Data preprocessing", the collected data are preprocessed based on methods and rules. Step 3 "Model training" the existing model is retrained based on the preprocessed data and historical data; this step takes place in parallel with the other steps. Step 4 "ML usage" the model that was trained in step 3 is used on the data that has already been cleaned up. Step 5 "Result" is the end of the pipeline. It shows the quality index that the machine learning model came up with.

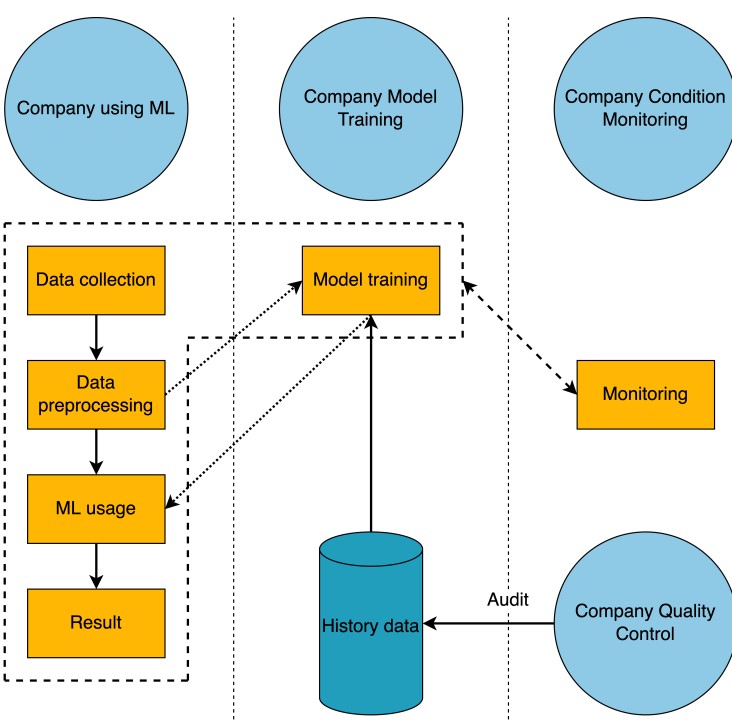

**Figure 5.** ML Pipeline and Stakeholder.

Figure 5 depicts four stakeholders who are often from different companies and are typically involved in the development and operation of a machine learning application for a SME as well as blockchain nodes (e.g., manufacturer, ML consultant, machine manufacturer, condition monitoring service provider, quality check auditor, etc.). For the goals to be met, the stakeholders must work together to manage their parts of the machine learning pipeline and affect the outcome. In the use case presented in Section 6.1, the actors are each from a different company. The actors are the company that is using ML, the company that is modelling training, the company that is conditioning monitoring, and the company that is quality control. However, stakeholders do not just influence parts of the pipeline; they influence each other by their reactions and also rate each other for trust value. A company that uses machine learning has control over and manages the following parts: data collection, data preprocessing, ML use, and results. So, these are the parts that make up the processing of data and the use of the ML model.

Company Model Training controls and manages the Model Training component and the certificate, which use machine learning and historical data to create a model from the company's collected and preprocessed data. *This is where the first mutual influence between stakeholders can be found: Company using ML and Company Model Training.*

Company Quality Control checks the history data to see which of the recorded images of the workpiece show a good surface. Quality control checks output data and assigns a trust level rating to it. *This is the second mutual influence that can be found among the actors: Company Quality Control, Company Model Training, Company using ML.*

Company Condition Monitoring manages and affects the monitoring of the following parts: data collection, data preprocessing, model usage, results of company ML use, and model training for company model training. *This is the third mutual influence between stakeholders that can be found: Company using ML, Company Model Training and Company Condition Monitoring.*

Lastly, it can be said that the people who have a stake in this pipeline can make decisions that affect the quality of the ML model, whether they are aware of it or not. Because of this, it is important that all stakeholders, all parts, and all data used to train the model can be tracked.

## 7. Evaluation

It is vital to construct the ML such that it is aware of the intricacies of the computing environment. To do this, ML frameworks will identify the maximum workload to be executed, which will require certain information such as memory capacity, processor speed, secured storage, and maybe additional capabilities such as multi-threading and secure communication channels. Then, from the most sensitive to the least sensitive, ML calculations are deployed into a pipeline that must be secure and specified as transparent to other stakeholders.

One essential trust increasing component is the "trust evaluation". This component allows to make overall decisions based on the trust level of the entity involved.

Protecting the whole pipeline is impossible without the engagement of many stakeholders in a trusting atmosphere. The proposed system can provide numerous trusted zones for more devices and, as a result, additional pipeline parts. To establish such "multi-party computation" based on blockchain, one must offer a verification mechanism for numerous participants, allowing stakeholders from other organisations to participate. For example, one certified pipeline might help in confirming the specifics and right setup, allowing stakeholders to cover many sites of the ML pipeline.

In addition to the pipeline workflow, selecting the most vulnerable areas of ML for security is not easy. On the most basic level, the proposed architecture provides a more trustworthy area in the ML pipeline, such as an additional trust base for experts or a model certificate. Furthermore, research on the privacy or integrity of pipeline components other than training and inference protection, such as data preparation, are very missing. Similar to training stage protection, such protection on a specific component of the pipeline will include threat (privacy and integrity) characterization, dataset certificate protection design, and performance evaluation. Following such work on future levels of the pipeline, full ML pipeline protection will be concretely developed and deployed to a greater degree and on a bigger scale. Table 11 lists countermeasures for potential ML pipeline attacks.

**Table 11.** Relevant Attacks and Countermeasures on the ML pipeline.

| Attacks on Pipeline | Countermeasures |
| --- | --- |
| Spoofing, Tampering on Data Collection | TLS is used by the system to read data from the server and to confirm the server's authenticity. Audit trails in the blockchain mitigates identity frauds and therefore data manipulation. |
| Tampering, Elevation of Privilege on Pre-Processing | Validate TLS use and data certificate verification. The blockchain ensures tamper resistance. |
| Tampering, Repudiation on ML Model | The model was distributedly stored on IPFS. Keeping the model's certificates and hash in the blockchain ensures tamper resistance. |

## 8. Conclusions

The safeguarding of AI-based solutions for the industry is essential to enabling trust in this new technology and allowing certification of products in the future, especially in sensitive manufacturing where ML is somehow involved in the producing process.

The use of blockchain as a framework to facilitate collaboration and authenticate and authorise stakeholders in the ML pipeline process is proposed in this paper. At the moment, changing the training data or the ML model to change the output or steal the ML model poses a security risk. By implementing our proposal for collaboratively registered stakeholders and using trust value for ML products and ML creators, we can reduce the risk of malicious data or codes when using a blockchain community certificate. A trust management system also aids in deterring malicious behaviour and encouraging more honest and qualified work from stakeholders. It can be shown that the proposed framework based on blockchain ensures data security and transparency in manufacturing for quality control and elaborates on the major benefits of the proposed approach, which are:

- a tamper-proof data history, which achieves accountability and supports quality audits;
- increases the trust by protocolling the used ML pipeline, by rating the experts involved in the ML pipeline and certifies for legitimacy for participation;
- certifies the pipeline infrastructure, ML model, data collection, and labeling.

In the evaluation section the mitigation of the security attacks threaten the framework have been demonstrated.

In future work, it can be possible to use benchmarking to evaluate by experts additionally on our proposal to improve the quality of ML certificates.

**Author Contributions:** Writing—original draft, F.S. and J.S.; Writing—review & editing, C.R. All authors have read and agreed to the published version of the manuscript.

**Funding:** This research was funded by the Federal Ministry of Education and Research (BMBF) under reference number COSMIC-X 02J21D144, and supervised by Projektträger Karlsruhe (PTKA).

**Institutional Review Board Statement:** Not applicable.

**Informed Consent Statement:** Not applicable.

**Acknowledgments:** The contents of this publication are taken from the research project "COSMIC-X-Kollaborative Smart Services für industrielle Wertschöpfungsnetze in GAIA-X", funded by the Federal Ministry of Education and Research (BMBF) under reference number COSMIC-X 02J21D144, and supervised by Projektträger Karlsruhe (PTKA). The responsibility for the content is with the authors.

**Conflicts of Interest:** The authors declare no conflict of interest. The funders had no role in the design of the study; in the collection, analyses, or interpretation of data; in the writing of the manuscript, or in the decision to publish the results.

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
