# Peer review of "Blockchain Secured Dynamic Machine Learning Pipeline for Manufacturing"

_applsci, doi:10.3390/app13020782_

Round 1

Reviewer 1 Report

The main research direction of this paper is the blockchain security dynamic machine learning pipeline for manufacturing industry. The authors first elaborate on the security risks faced by ML and propose a blockchain-based framework to ensure data security and transparency for quality control in the manufacturing process. And the participating entities are rated, certifying their legitimacy, thereby increasing the credibility of the ML pipeline.

The paper is well-organized, easy to understand. In my view this paper can be accepted, but there are some minor problems that need to be revised:

-            The title of the article is "Blockchain Secure Dynamic Machine Learning Pipelines for Manufacturing". The article should mention more about "Manufacturing" to make the content more closely linked.

-            Figure 3 is a system framework diagram, but this diagram is too simple, readers cannot get useful information. It is recommended that the author modify the image.

-            When describing the structure of the article in the Introduction, why do you need to describe Secsection2.1 separately? I don't think this is very necessary. It is more suitable to directly write the content of Section2 here.

-            The article has many formatting errors. For example, in Page11, the capitalization of the first letter after Step is problematic.

-            To broaden the scope of this paper, the authors should refer to some paper such as: Data query mechanism based on hash computing power of blockchain in Internet of Things; Novel vote scheme for decision-making feedback based on blockchain in internet of vehicles; Multiple cloud storage mechanism based on blockchain in smart homes.

Author Response

We appreciate the time and effort that you have dedicated to providing your valuable feedback on our manuscript. We are grateful for the insightful comments on our paper. Furthermore, we have been able to incorporate changes to reflect most of the suggestions provided. We have highlighted the changes within the manuscript. Here is a point-by-point response to the comments and concerns.

Comment 1: The article should mention more about "Manufacturing" to make the content more closely linked.

Response: Thank you for your comment. We try to include more references to manufacturing in use cases and intrusions. 

Comment 2: Figure 3 is a system framework diagram, but this diagram is too simple, readers cannot get useful information. It is recommended that the author modify the image.

Response: We changed it to show more detail about the suggested method.

Comment 3: When describing the structure of the article in the Introduction, why do you need to describe Secsection2.1 separately? I don't think this is very necessary. It is more suitable to directly write the content of Section2 here.

Response: We did a major change in introduction section based on your comment.

Comment 4: The article has many formatting errors. For example, in Page11, the capitalization of the first letter after Step is problematic.

Response: Thank you for taking the time to notice us. We reviewed the paper and made structural and grammatical corrections. 

Comment 5: To broaden the scope of this paper, the authors should refer to some paper such as: Data query mechanism based on hash computing power of blockchain in Internet of Things; Novel vote scheme for decision-making feedback based on blockchain in internet of vehicles; Multiple cloud storage mechanism based on blockchain in smart homes

Response: Thank you for recommending papers; we have added them as references 29-31.

Reviewer 2 Report

The article is rejected as it did not sufficiently meet the quality standards of the journal. It should be strengthened both in terms of methodology and literature analysis (this would benefit the soundness of the work), and in the validation phase of the proposed framework.  Furthermore, the English language should also be improved; there are several grammatical oversights in the article and the logical flow of exposition of the arguments is sometimes lacking.

I therefore suggest that there should be a section dedicated to the research method adopted so that the entire research process and the tools used are clear. Furthermore, greater emphasis should be placed on the analysis of the state of the art as an indispensable basis for the research conducted. Finally, the proposed framework should be validated in one or more relevant industrial cases.

Author Response

We appreciate the time and effort that you have dedicated to providing your valuable feedback on our manuscript. We are grateful for the insightful comments on our paper. Furthermore, we have been able to incorporate changes to reflect most of the suggestions provided. We have highlighted the changes within the manuscript. Here is a point-by-point response to the comments and concerns.

Comment 1: I therefore suggest that there should be a section dedicated to the research method adopted so that the entire research process and the tools used are clear.

Response: Thank you for your comment to improve the quality of our paper. We added this section to our paper.

Comment 2: Furthermore, greater emphasis should be placed on the analysis of the state of the art as an indispensable basis for the research conducted.

Response: We changed the state of art and security analysis to address your concern.

Comment 3: Finally, the proposed framework should be validated in one or more relevant industrial cases.

Response: To make our work clear and make sense in our imagination, we added an industrial use case after the proposed approach.

Reviewer 3 Report

 Although the topic of the article is interesting, it is rejected because it lacks scientific and methodological rigour. Furthermore, the entire paper should be improved in terms of both the logical flow of the content and the English language. In the following, I provide some comments that can help authors improve the quality of the manuscript.

ABSTRACT

·       The logical flow of content should be improved in order to allow for a better understanding. Moreover, the innovative contribution of the work (both from an academic and managerial point of view) should be made clearer.

 INTRODUCTION

·      It is necessary to increase the number of bibliographical references to corroborate the information made (e.g. between lines 26 and 40).

 ·      It would be important to explicitly bring out the correlation between ML, AI and cybersecurity and blockchain.

 ·       The logical flow between the various paragraphs needs to be improved. Often, a topic is interrupted and then addressed a few lines later.

 ·       No clear research gap emerges that the study intends to fill

 ·       The proposed blockchain-based framework does not emerge on what methodological basis it is based

 The article does not define the research method adopted to achieve the research objectives

 Section 2 'Machine Learning Application in Manufacturing'

·       I would move it after the section on the state of the art. In any case, this section should also be improved both in terms of conceptual flow and from a quality point of view.

 Section 3 'State of the art”

·       The section on the state of the art should be enhanced and better structured. It could be written at the beginning of the section what is being researched in the literature to clarify things for the reader straight away. Lastly, a more accurate comparative analysis of what is emerging would be appropriate.

 Section 4

·       It should be enhanced in terms of bibliographical references and the summary table should be better structured (both by enriching the content and by inserting appropriate bibliographical references)

 Section 5

·       The contents of Section 5 'Proposed method for mitigation' should be validated in a case study and properly supported by the literature

Section 7

·       The conclusions are lacking in content and soundness. There is no reference to any academic benefit of the study conducted. Furthermore, little emphasis is placed on future works. Finally, the field of medical technology production is mentioned in this section, as well as in the abstract, but there are no other parts within the manuscript where reference is made to it.

Author Response

We appreciate the time and effort that you have dedicated to providing your valuable feedback on our manuscript. We are grateful for the insightful comments on our paper. Furthermore, we have been able to incorporate changes to reflect most of the suggestions provided. We have highlighted the changes within the manuscript. Here is a point-by-point response to the comments and concerns.

Comment 1: The logical flow of content should be improved in order to allow for a better understanding. Moreover, the innovative contribution of the work (both from an academic and managerial point of view) should be made clearer.

Response: Thank you for your comment. We changed it based on your comment to improve the abstract

Comment 2: It is necessary to increase the number of bibliographical references to corroborate the information made (e.g. between lines 26 and 40).

Response: Thank you for your comment. We added more references to that part.

Comment 3: It would be important to explicitly bring out the correlation between ML, AI and cybersecurity and blockchain.

Response: It is added in first paragraph of introduction.

Comment 4,5,6: The logical flow between the various paragraphs needs to be improved. Often, a topic is interrupted and then addressed a few lines later.

 ·       No clear research gap emerges that the study intends to fill

 ·       The proposed blockchain-based framework does not emerge on what methodological basis it is based

Response: Thank you for taking the time to notice us. We reviewed the paper and made structural and grammatical corrections on introduction

Comment 5: The article does not define the research method adopted to achieve the research objectives

Response: we added the research methodology section and change the structure of related section to answer your concerns. 

Comment 6,7:  I would move it after the section on the state of the art. In any case, this section should also be improved both in terms of conceptual flow and from a quality point of view.

The section on the state of the art should be enhanced and better structured. It could be written at the beginning of the section what is being researched in the literature to clarify things for the reader straight away. Lastly, a more accurate comparative analysis of what is emerging would be appropriate.

Response: we moved it and change the State of art to satisfy the concerns.

Comment 8:  It should be enhanced in terms of bibliographical references and the summary table should be better structured (both by enriching the content and by inserting appropriate bibliographical references)

Response: We made the changes requested in the comment. 

Comment 9:   The contents of Section 5 'Proposed method for mitigation' should be validated in a case study and properly supported by the literature

Response: Following the mitigation section, we added a use case. 

Comment 10:    The conclusions are lacking in content and soundness. There is no reference to any academic benefit of the study conducted. Furthermore, little emphasis is placed on future works. Finally, the field of medical technology production is mentioned in this section, as well as in the abstract, but there are no other parts within the manuscript where reference is made to it.

Response: We rewrote the conclusion in response to your comment.

Round 2

Reviewer 2 Report

The quality of the manuscript was greatly improved, based on the comments received. I therefore believe that, after minor further revisions, the paper can be published.

In particular, it would be necessary to improve the form of exposition of the contents of the abstract and the introduction as some sentences could be integrated into each other. This would make for lighter reading.

Author Response

Thank you for your review, please see the attachment

Reviewer 3 Report

The article has been considerably improved in line with the revisions received. Only a few minor revisions are necessary before publication:

1.     Although the content of the abstract has been improved, it should be streamlined. For example, the phrases "Since ML pipelines are very complex..." and "Since different and chancing stakeholders..." can be integrated with each other; moreover, many sentences begin with "This..." . The sentence "This paper provides an overview...transparency in ML pipelines" needs a conjunction.

2.       Insert the extended version of AI the first time you mention it

3.     The introduction was also improved according to the suggestions received. However, it should be lightened in some places by integrating a few sentences. The final part describing the structure of the article could be improved in the form of exposition.

4.       line 93 - define -> definition of

5.       line 412: In Figure 4 -> Figure 4 shows

Author Response

Thank you for your review, please see the attachment.
